# Microbiome Markers of Pancreatic Cancer Based on Bacteria-Derived Extracellular Vesicles Acquired from Blood Samples: A Retrospective Propensity Score Matching Analysis

**DOI:** 10.3390/biology10030219

**Published:** 2021-03-13

**Authors:** Jae Ri Kim, Kyulhee Han, Youngmin Han, Nayeon Kang, Tae-Seop Shin, Hyeon Ju Park, Hongbeom Kim, Wooil Kwon, Seungyeoun Lee, Yoon-Keun Kim, Taesung Park, Jin-Young Jang

**Affiliations:** 1Department of Surgery, Seoul National University Hospital, Seoul 03080, Korea; jaripo@gmail.com (J.R.K.); vickijoa@naver.com (Y.H.); surgeonkhb@gmail.com (H.K.); willdoc78@gmail.com (W.K.); 2Department of Surgery, Gyeongsang National University Changwon Hospital, Changwon 51472, Korea; 3Interdisciplinary Program in Bioinformatics, Seoul National University, Seoul 08826, Korea; hgh97617@gmail.com (K.H.); thisisnykang@gmail.com (N.K.); 4MD Healthcare Inc., Seoul 03923, Korea; tsshin@mdhc.kr (T.-S.S.); hjpark@mdhc.kr (H.J.P.); ykkim@mdhc.kr (Y.-K.K.); 5Department of Mathematics and Statistics, Sejong University, Seoul 05006, Korea; leesy@sejong.ac.kr; 6Department of Statistics, Seoul National University, Seoul 08826, Korea

**Keywords:** pancreatic cancer, microbial extracellular vesicles, microbiome markers, early diagnosis, propensity score matching

## Abstract

**Simple Summary:**

Although tremendous advances in diagnosis and treatment, pancreatic cancer still remains one of the lethal diseases with an overall survival rate of 10~15%. Early detection and diagnosis of pancreatic cancer is very important in improving the prognosis of patients. The aim of our study was to find new biomarkers, using microbiomes based on bacteria-derived extracellular vesicles, extracted from blood serum. With 38 patients with pancreatic cancer and 52 healthy controls with no history of pancreatic disease, we identified several compositional differences of microbiome between them. Using various combinations of the metagenomic markers which made the compositional differences, we also built a pancreatic cancer prediction model with high area under the receiver operating characteristic curve (0.966 at the phylum level and 1.000 at the genus level). These microbiome markers, based on bacteria-derived extracellular vesicles acquired from blood, show demonstrate the potential of candidate biomarkers for early diagnosis of pancreatic cancer.

**Abstract:**

Novel biomarkers for early diagnosis of pancreatic cancer (PC) are necessary to improve prognosis. We aimed to discover candidate biomarkers by identifying compositional differences of microbiome between patients with PC (n = 38) and healthy controls (n = 52), using microbial extracellular vesicles (EVs) acquired from blood samples. Composition analysis was performed using 16S rRNA gene analysis and bacteria-derived EVs. Statistically significant differences in microbial compositions were used to construct PC prediction models after propensity score matching analysis to reduce other possible biases. Between-group differences in microbial compositions were identified at the phylum and genus levels. At the phylum level, three species (*Verrucomicrobia*, *Deferribacteres*, and *Bacteroidetes*) were more abundant and one species (*Actinobacteria*) was less abundant in PC patients. At the genus level, four species (*Stenotrophomonas*, *Sphingomonas*, *Propionibacterium*, and *Corynebacterium*) were less abundant and six species (*Ruminococcaceae UCG-014*, *Lachnospiraceae NK4A136 group*, *Akkermansia*, *Turicibacter*, *Ruminiclostridium*, and *Lachnospiraceae UCG-001*) were more abundant in PC patients. Using the best combination of these microbiome markers, we constructed a PC prediction model that yielded a high area under the receiver operating characteristic curve (0.966 and 1.000, at the phylum and genus level, respectively). These microbiome markers, which altered microbial compositions, are therefore candidate biomarkers for early diagnosis of PC.

## 1. Introduction

Pancreatic cancer (PC) is the leading cause of cancer-associated mortality worldwide [1]. Despite advances in diagnosis, surgical technique, perioperative care, and chemoradiotherapy regimens, the 5-year survival rate for PC patients remains 10–15% due to difficulties with early detection. Only 20% of PC patients are considered to have “early-stage” PC at the time of diagnosis, when curative-intent surgery is possible [2]. Research seeking to discover novel biomarkers for early diagnosis is on-going.

To date, several studies have reported that dysbiosis of the human body microenvironment can increase the risk of inflammation or malignant tumor development [3]. The gut microbiota has been reported to affect the risk of adverse health outcomes, including that of liver cirrhosis/hepatocellular carcinoma [4,5], cardiovascular disease [6], breast cancer [7], and colorectal cancer [8]. Moreover, previous studies have suggested an association between PC and the presence of gut bacteria, such as *Helicobacter pylori*, and salivary microbiota have been shown to be associated with chronic pancreatitis or PC [9,10,11]. However, alterations of blood microbiota in patients with PC remain unknown.

Microbiota produce large quantities of bacteria-derived extracellular vesicles (EVs), which allow interaction with human cells [12]. EVs contain bacteria-derived genomes and are detectable in human blood, urine, bile, and stool samples. They can help evaluate microbiota composition [13,14,15]. A recent study, aimed to develop a diagnostic model for ovarian cancer, used microbiome profiles from serum bacteria-derived EVs alongside clinicopathologic data [16]. However, to the best of our knowledge, the association between the blood microbiome, assessed with bacteria-derived EVs, and PC has not been previously studied. This is the first study to identify candidate biomarkers for the diagnosis of early-stage PC, by comparing the differences in microbiome compositions using blood microbiota between patients with PC and healthy controls.

## 2. Results

After quality control of samples, 82 patients with PC and 116 healthy controls were identified as eligible for this study. However, the age and sex of the two groups were different enough to discriminate PC patients from controls with a high area under the ROC curve (AUC) of 0.882. After propensity score matching (PSM), only 38 patients with PC (17 men and 21 women) and 52 controls (17 men and 35 women) were selected. As a result, the covariates could no longer contribute to predicting PC (AUC of 0.571, Appendix A
Figure A1).

The age and sex of the patients before and after PSM are described in Table 1A. Table 1B shows the clinicopathologic characteristics of the PC group after PSM. Most PC patients were pathologically confirmed as having advanced stage cancer (68.4% had stage II and 29.0% had stage III/IV, as defined by the seventh American Joint Committee on Cancer). Although all 38 patients underwent surgery, 6 of them received palliative-intent surgery owing to inoperable primary lesions or metastases identified during the procedure. Two patients had histological findings other than adenocarcinoma, including one patient with colloid carcinoma and another with adenosquamous cell carcinoma.

### 2.1. Microbiome Composition Comparisons

Using 21 phyla and 353 genera based on 90 samples, we assigned 1118 and 978 operational taxonomy units (OTUs) at the phylum and genus levels, respectively. The number of OTUs was 789 and 850 in the PC and control groups, respectively. Figure 1a shows between-group differences in α-diversity (within-sample diversity) based on the Shannon index. Although there were no significant differences at the genus level (*p* = 0.14, Wilcoxon rank-sum test), the median Shannon index was higher for the PC than for the control group at the phylum level (*p* = 0.0084, Wilcoxon rank-sum test). An overview of each level’s composition is shown in Figure 1b. Figure 2 shows the multidimensional scaling plots at the OTU and genus levels, which capture β-diversity (between-sample variability). principal coordinate analysis (PCoA) based on the Aitchison and Bray-Curtis distances revealed dot patterns, suggesting that the PC patients and controls were distinct at both levels.

### 2.2. Biomarker Selection

Depending on the statistical method that was used, between-group differences in microbial composition at the phylum level included *Verrucomicrobia* (significant difference by nine methods), followed by *Deferribacteres* and *Actinobacteria* (significant by seven methods), and *Bacteroidetes* (significant by six methods) (Table 2). At the genus level, using more than six methods, differences in the following communities were found: *Stenotrophomonas*/*Sphingomonas*/*Ruminococcaceae UCG-014*/*Propionibacterium*/*Lachnospiraceae NK4A136 group* (significant by eight methods), *Akkermansia* (significant by seven methods), and *Turicibacter*/*Ruminiclostridium*/*Lachnospiraceae UCG-001*/*Corynebacterium* (significant by six methods). Figure 3 shows the overall log2 counts of each marker, selected through statistical methods for the PC and control groups. At the phylum level, the abundance of *Verrucomicrobia*/*Deferribacteres*/*Bacteroidetes* was more and that of *Actinobacteria* was less in the PC group than in the control group. Moreover, the abundance of *Stenotrophomonas*/*Sphingomonas*/*Propionibacterium*/*Corynebacterium* was decreased and that of the other six microbiota increased in the PC compared with the control group at the genus level.

### 2.3. Development of PC Prediction Model

We developed prediction models using statistically significant microbiome markers (Figure A2). After an exhaustive search that considered all possible combinations using randomly separated model development (MD) and test sets, we arrived at the best models for the phylum and genus levels, respectively (Table A1). Figure 4 shows the overall performance of the model. The selected markers in the best fitting models were *Verrucomicrobia* and *Actinobacteria* at the phylum level and *Sphingomonas*, *Ruminococcaceae UCG-014*, *Propionibacterium*, *Akkermansia*, *Ruminiclostridium*, *Lachnospiraceae UCG-001*, and *Corynebacterium 1* at the genus level. Sensitivity, specificity, and accuracy estimates were average values calculated by 2-fold cross validation. In the test set, the AUC of these best models was 0.966 at the phylum level and 0.913 at the genus level. The receiver operating characteristic (ROC) curves of these models are shown in Figure 5.

### 2.4. In Vitro Experiments to Determine the Biological Functions of Bacteria-Derived EVs

As a previous result, we found several microbiome markers that decreased from the PC patients. From this, to elucidate the role of decreased markers from PC patients, we generated and isolated the EV from *C. glutamicum* culture medium that almost same the selected marker bacteria *Corynebacterium 1*. Transmission electron microscopy image (Figure 6a), Dynamic light scattering (Figure 6b) and nanoparticle tracking analysis (Figure 6c) show that the EV from *C. glutamicum* was averagely 133. 3 nm size and the 7.85 × 10^11^ particles were in the 1 mg/mL of the samples. Sodium dodecyl sulfate-polyacrylamide gel electrophoresis was performed to determine protein patterns and protein size distribution (Figure 6d). Furthermore, to evaluate its functional efficacy to inhibit expression of tumor necrosis factor-α (TNF-α), which related the carcinogenesis, we used an EV obtained from *E. coli* as a stimulant to trigger the secretion of TNF-α. As a result, the amount of secreted TNF-α was reduced in all samples prepared with *C. glutamicum* EV at a concentration of 0.1 to 10 μg/mL (Figure 6e). The amount of TNF-α showed a tendency to decrease depending on the concentration of *C. glutamicum* EV.

### 2.5. Sensitivity Analysis According to Various Matching Conditions

At the phylum level, the markers (*Verrucomicrobia* and *Actinobacteria*) that were used in the final prediction models were consistently selected regardless of the changes in the calipers (Figure A3). However, among the seven markers used at the genus level, only four (*Akkermansia*, *Sphingomonas*, *Ruminococcaceae UCG-014*, and *Propionibacterium*) were selected consistently according to the changes in the calipers.

To identify the performance of the prediction models under the various calipers, we also analyzed the AUCs of the other best models that were independently selected in each condition of changed calipers, in addition to investigating the changes of the AUCs of the selected best model (with 2 markers at the phylum level and 7 at the genus level) found in this study (Figure A4). As a result, under various conditions, the testing AUCs of the microbial marker models were quite constant compared with those of the covariate model.

## 3. Discussion

This is the first study to investigate the altered composition of microbiomes in patients with PC and to evaluate the relationship between microbiota and PC, using bacteria-derived EVs. We performed 16S rRNA gene analysis and compared microbiome composition between patients with PC and healthy controls. Some of the compositional differences identified in this study might be candidate biomarkers of early-stage PC; using these candidate biomarkers, we proposed and validated a PC prediction model.

Previous studies have reported on the relationship between microbiota and PC or chronic pancreatitis. Microbial composition in patients with PC is known to be altered at several sites, including the oral cavity, gastrointestinal tract, and pancreatic tissues. Microbial risk factors for PC include oral microbiota in periodontal disease, altered presence of *H. pylori*, and hepatotropic viruses. Periodontitis is a form of chronic gingival inflammation and a common type of oral infection, previously linked to the risk of pancreatic [11,17,18,19,20] and other organ [8,21,22] malignancies. Several pathogens, including *P. gingivalis*, *N. elongata*, *Fusobacterium*, and *S. mitis* have been reported. Evading the host immune system, these pathogens can trigger the Toll-like receptor signaling pathways and promote pancreatic carcinogenesis in animal models [23].

*H. pylori* is a gut microbe that can reach the pancreas through the circulatory system or the pancreatic/biliary duct. However, the relationship between *H. pylori* and PC remains controversial. To date, it has been suggested that this relationship is mediated by specific risk factors, including infection with cytotoxin-associated gene A-negative *H. pylori* strain [24], non-O blood type [25], and smoking [26]. Moreover, hepatotropic viruses, including hepatitis B virus and hepatitis C virus, have been associated with direct and indirect (via pancreatitis) development of PC [27,28,29].

Previous studies of microbiota composition used fecal, salivary, biliary, or tissue samples. Findings obtained from samples collected of the digestive system might reflect the genome profiles of gut microbiota. Meanwhile, several recent studies reported on the presence of EVs that contain bacterial genome DNA fragments in serum. The size of EVs from either gram-negative or gram-positive bacteria are very small (10 to 300 nm in diameter) [30,31], allowing them to cross intestinal cellular membrane and travel throughout the blood system [32]. Based on these findings, subsequent microbiome analyses using bacteria-derived EVs extracted from serum were undertaken [5,33].

In the present study, at the phylum level, *Actinobacteria* were less and *Verrucomicrobia* were more abundant in the PC group than in control group. At the genus level, *Akkermansia*/*Ruminococcaceae UCG-014*/*Ruminiclostridium* were more abundant and *Sphingomonas*/*Propionibacterium*/*Corynebacterium 1* were less abundant in PC patients than in the control group. Some of these findings are consistent with those from previous studies, which involved fecal microbiome analysis of PC patients [34].

Some species of *Actinobacteria* are known to produce butyrate and modulate immune function. Reduction of butyrate levels can promote inflammation, which acts as an anti-inflammatory agent mainly by blocking the activation of nuclear factor kB in intestinal epithelial cells [35]. Consistent with our findings, abundance of *Actinobacteria* in colorectal cancer patients has been reported to be lower than that in the control group [36]. In addition, while the exact mechanism remains unclear, *Akkermansia* (in *Verrucomicrobia* phylum) is known as an immune modulator, likely related to the programmed cell death protein 1 blockade pathway [37]. Finally, some *Sphingomonas* species are known to have the ability to stimulate natural killer T cells, which suppress tumor progression [38]. However, the specific function and exact mechanism associated with carcinogenesis in most taxa remain poorly understood. Future preclinical research is necessary to understand the relationship between these microbiota and PC.

As stated, few previous studies have evaluated bacteria-derived EVs acquired from blood samples; our report is the first such study on PC patients. Although we found some differences in the microbiome profiles of PC patients, we were unable to determine their biological function or behavior. Future metagenomic research should endeavor to elucidate the role of microbiota in the transition from normal to malignant tissue. In in vitro experiments, we found that the amount of acute phase inflammatory cytokines tended to decrease depending on the concentration of *C. glutamicum* EV. Therefore, for PC patients, a low level of *C. glutamicum* might be associated with inflammation, which can trigger cancer development.

Alongside its novelty, this study has some limitations. First, the number of samples was relatively small. We performed PSM and sensitivity analysis with various calipers to increase the generalizability of our findings. Second, although we performed internal validation with randomly separated MD and test sets, external validation with a larger cohort is required. Third, we could not explain the composition of gut microbiota directly through this study. Bacteria-derived EVs present in blood samples are believed to mostly originate from the gut microbiota. However, factors such as gut barrier, host immunity, and organ status can alter the composition of microbiota detected in the blood, making it distinct from that concurrently present in the gut. Further studies are required to describe and explain these suspected differences and their implications, alongside the functional interchangeability of gut and blood microbiota.

## 4. Materials and Methods

This study included patients diagnosed with PC between 2009 and 2015 at the Seoul National University Hospital and healthy controls who received regular checkups at the Seoul National University Boramae Hospital and Inje University Haeundae Paik Hospital. All patients with PC underwent surgical resection, and final pathology reports were confirmed. The control group included healthy adults without any clinical or imaging evidence of pancreatic disease or history of other cancers. Following data collection, including blood samples, the final study population was selected based on PSM analysis to reduce selection bias by equating the groups based on the covariates. This study complied with the principles of the Declaration of Helsinki and was approved by the institutional review board of Seoul National University Hospital (1601-137-739). The informed consent requirement was waived due to the retrospective nature of the study and use of anonymous clinical data.

### 4.1. Blood Sample Preparation and DNA Extraction

The blood samples were placed into Vacutainer EDTA tubes, and serum was centrifuged at 2000× *g* for 15 min at 4 °C to remove cell debris. The supernatant was collected and incubated with proteinase K at 56 °C for 30 min. Subsequently, the samples were boiled at 100 °C for 40 min to extract DNA from EVs; afterward, the supernatant was collected by centrifugation at 10,000× *g* at 4 °C. A DNA isolation kit (DNeasy Blood & Tissue Kit, QIAGEN, Hilden, Germany) was used to extract the total DNA from 1 mL of supernatant. The quality and quantity of DNA were measured using the QIAxpert system (QIAGEN, Hilden, Germany).

### 4.2. Microbiomic Sequencing

Bacterial genomic DNA was amplified with 16S_V3_F (5-TCGTCGGCAGCGTCAGATGTGTATAAGAGACAGCCTACGGGNGGCWGCAG-3) and 16S_V4_R (5-GTCTCGTGGGCTCGGAGATGTGTATAAGAGACAGGACTACHVGGGTATCTAATCC-3) primers, which are specific to V3-V4 hypervariable regions of the 16S rDNA gene [39]. The libraries were prepared using polymerase chain reaction products according to the MiSeq System guide (Illumina, San Diego, CA, USA) and quantified using QIAxpert (QIAGEN, Hilden, Germany). Each amplicon was then quantified and sequenced on MiSeq (Illumina, San Diego, CA, USA) according to the manufacturer’s recommendations.

### 4.3. Taxonomic Assignment and Profiling

Raw pyrosequencing reads from the sequencer were filtered using MiSeq (Illumina, SanDiego, CA, USA), according to the barcode and primer sequences. Taxonomic assignment was performed by the profiling program MDx-Pro ver.1 (MD Healthcare, Seoul, Korea). After checking the read length (≥300 bp) and quality score (average Phred score ≥ 20), high-quality sequencing reads were selected. OTUs were clustered using sequence clustering algorithms CD-HIT [40].

Subsequently, taxonomic assignment was performed using UCLUST and QIIME against the GREENGENES reference database (gg_13_5_99) [41,42]. OTUs with sequences < 0.005% of the total were removed from the OTU table; a total of 1134 OTUs were obtained. Samples with a low number of read counts (<2500) were filtered during quality control process. The resulting OTU table was used for predictive functional analysis with Tax4Fun software (metagenomics package version 0.1.014) [43].

### 4.4. Propensity Score Matching and Statistical Analysis

We used PSM analysis to minimize the impact of covariates on effect estimates. The propensity score is a probability that a unit with specific characteristic will be assigned to treatment group. PSM is a statistical matching technique that uses the propensity score to estimate the effectiveness of interventions, given particular covariates [44]. For example, PSM was used to adjust non-random drug assignment to determine whether drugs have the effects of protecting infants from apnea [45]. PSM was also applied to reduce the selection bias for estimating breast cancer risk in relation to antidepressant medications [46]. While PSM has been used for the case of large samples, it was shown by a simulation study that PSM performed well for the case of small samples [47]. Recently, PSM was successfully applied to fecal microbiota studies. PSM reduced the influence of lifestyle variables which might attenuate the relevance between fecal bacteria and the risk of gastric cancer [48]. PSM was also used to control the effect of clinical variables on microbiota composition to find the relationship between fecal microbiota and Parkinson’s disease [49].

We applied PSM to our case-control data to reduce heterogeneity of age and sex between PC and control groups. We considered age and gender as potential confounders, because those were highly unbalanced between PC and control groups. To evaluate the consistency of estimates under varying matching conditions, sensitivity analysis was performed at various levels of matching, using calipers [50], which are values that express the strictness of covariate matching for a given propensity score. Small caliper values represent strict matching; in contrast, large caliper values represent models that are close to those observed with random sampling. We measured trends of the selected markers and their performances and built prediction models according to the change in calipers.

All statistical analyses of clinical data were performed in R version 4.0.2 on Windows 10 (Version 4.0.2, http://www.R-project.org (accessed on 15 December 2020)). Categorical variables were presented as counts with percentages and compared using the chi-square test. Continuous variables were compared using a Kruskal-Wallis test or one-way analysis of variance (ANOVA). A *p*-value < 0.05 was considered indicative of a statistically significant difference.

The α-diversity of microbiota for each sample was evaluated using the Shannon index. The Wilcoxon rank-sum test was used to compare α-diversities between groups. Moreover, β-diversity was measured by Aitchison and Bray-Curtis distances with 90 matched samples [51,52]. Based on these distance measures, PCoA was performed, and the cmdscale function in R was used to assess the statistical significance of separation among groups. When comparing groups and constructing models using calculated relative abundance, OTUs were used without those uncultured or unidentified. However, uncultured or unidentified OTUs were included in the analysis to provide an overview of microbial composition in each group.

### 4.5. Marker Selection and Prediction Model Development for PC

Using nine statistical methods, including microbiome-specific methods (metagenomeSeq [Gaussian, Log Normal], ZIBSeq, ANCOM, CLR permutation), simple statistical test (Wilcoxon rank-sum test), and conventional methods for identifying differentially expressed genes in RNAseq data (DESeq2 [LRT, Wald], edgeR), microbiome was sorted in order of compositional proportion. The relative differences in OTUs abundances between the PC and control group were investigated to identify candidate cancer biomarkers. We selected the candidate OTUs with an average relative abundance of >1% and *p*-value < 0.05.

To develop a prediction model for PC, we randomly divided our samples into MD and test sets to minimize selection bias. Logistic regression analysis was performed using selected OTUs via an exhaustive search method. All possible combinations of candidate OTUs were tested by repeating 10 times for 2-fold cross validation to find the optimal variable combination to discriminate between the PC and control group. The final model was selected based on the lowest Akaike’s information criteria from among the development set; it was subsequently validated using ROC curves and the AUC calculation based on the test set [53,54].

### 4.6. Additional In Vitro Experiments Using Corynebacterium glutamicum Strain

To identify the basic characteristics of bacteria-derived EVs, we performed additional in vitro experiments using *Corynebacterium glutamicum* strain (which was the same as the *Corynebacterium 1*, reported at the genus level), which was less in abundance in PC patients than in the controls. *C. glutamicum* KCTC 9097 was cultured in LB medium for 15 h at 37 °C. The cultured solution was collected and centrifuged at 10,000× *g* for 15 min. After filtration (0.22 µm) of the supernatant separated from the cells, EVs were extracted using ultrafiltration and ultracentrifugation methods. Five microliters of diluted EVs were dropped on Formvar-carbon coated EM grids and left aside to allow membranes adsorb for 2 min. The vesicle-coat grids were fixed with 0.25% glutaraldehyde for four minutes and washed twice with distilled water for one minute each. The grids were stained with 2% uranyl acetate at pH 7 for 5 min and viewed using a H-7650 transmission electron microscope (Hitachi, Tokyo, Japan) at a voltage of 80 kV. To identify the size and distribution of the particles of *C. glutamicum* EV, DLS and NTA were performed.

## 5. Conclusions

This study revealed compositional differences of microbiome between patients with PC and healthy controls, following covariate matching that reduced the impact of selection bias. Among altered microbial communities, we identified candidate biomarkers such as *Verrucomicrobia* and *Actinobacteria* at the phylum and *Sphingomonas*, *Ruminococcaceae UCG-014*, *Propiobacterium*, *Akkermansia*, *Ruminiclostridium*, *Lachnospiraceae UCG-001*, and *Corynebacterium* at the genus level, while developing prediction models for PC. Further studies with larger cohorts are necessary to validate the present findings. Moreover, research is required into rare microbial strains whose roles in the host immune system function or in carcinogenesis remains unclear.

## Figures and Tables

**Figure 1 biology-10-00219-f001:**
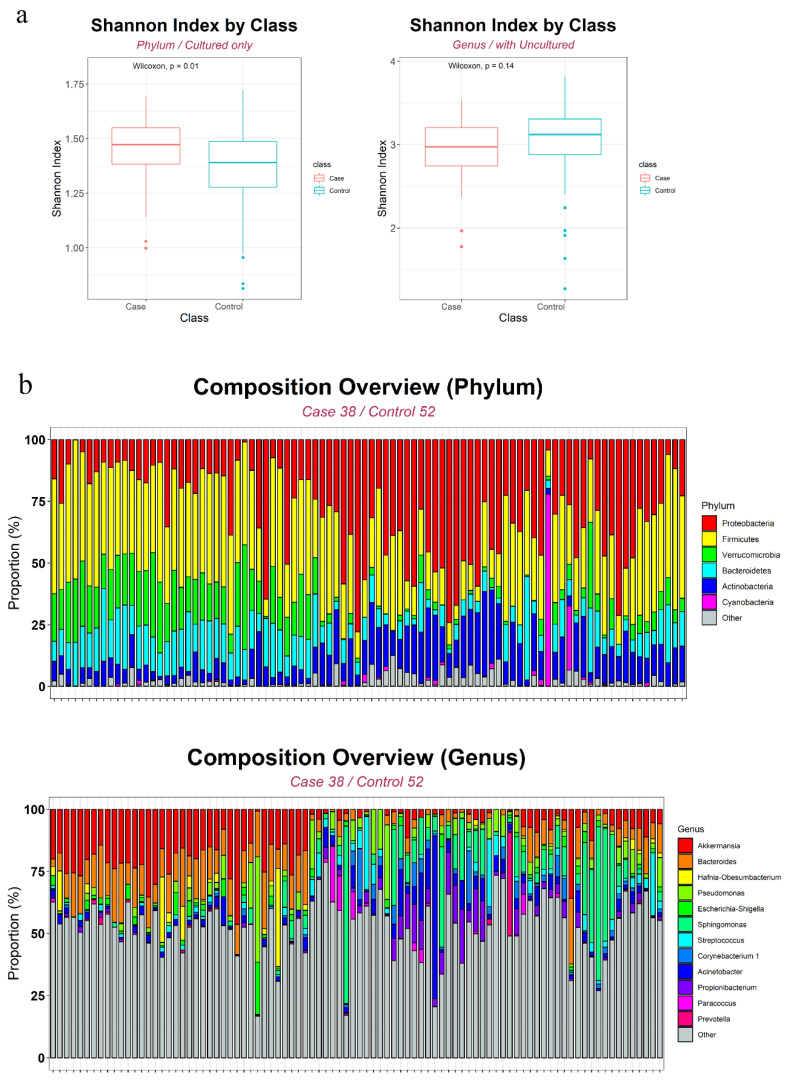
Alpha diversity (**a**) and overall microbiome composition analysis at phylum and genus level (**b**).

**Figure 2 biology-10-00219-f002:**
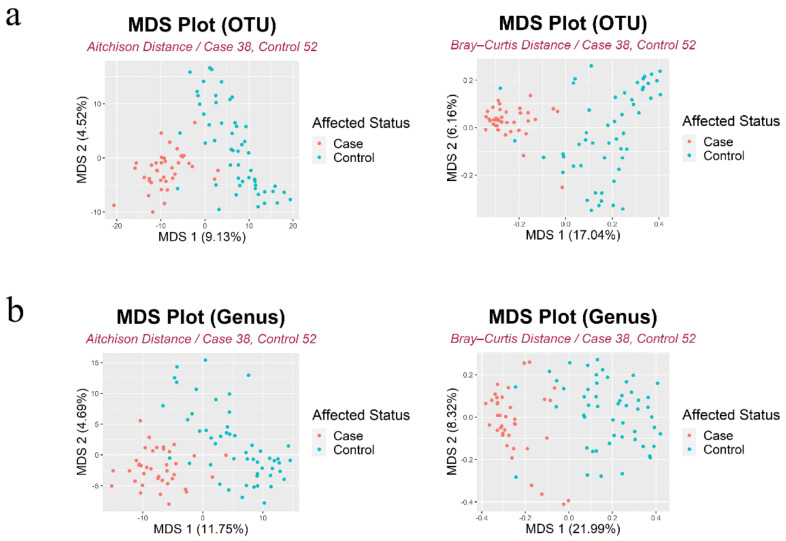
The beta diversity using multidimensional scaling (MDS) plot at OTU (**a**) and genus level (**b**).

**Figure 3 biology-10-00219-f003:**
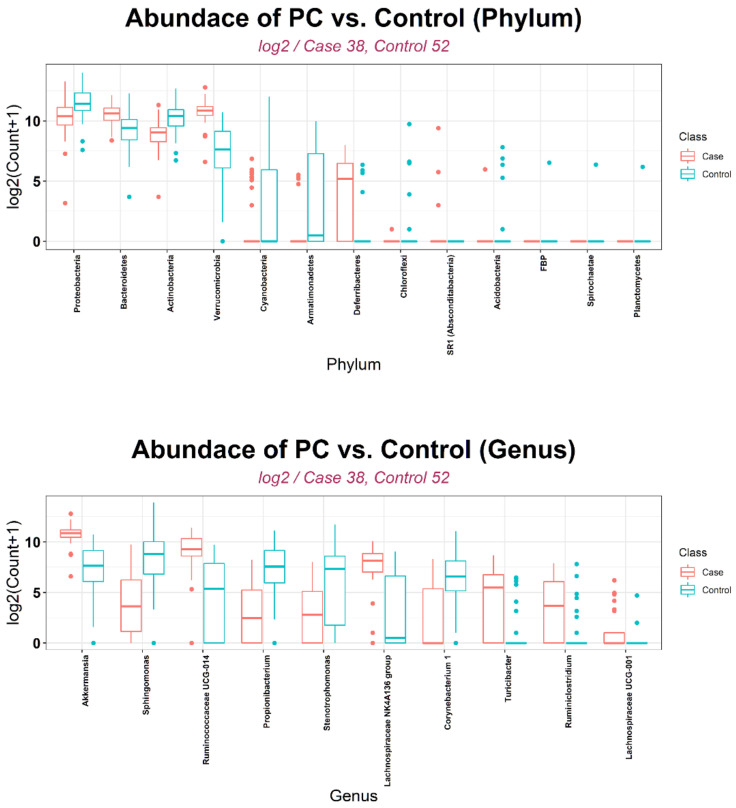
Log2 counts of abundant OTUs in PC and control group.

**Figure 4 biology-10-00219-f004:**
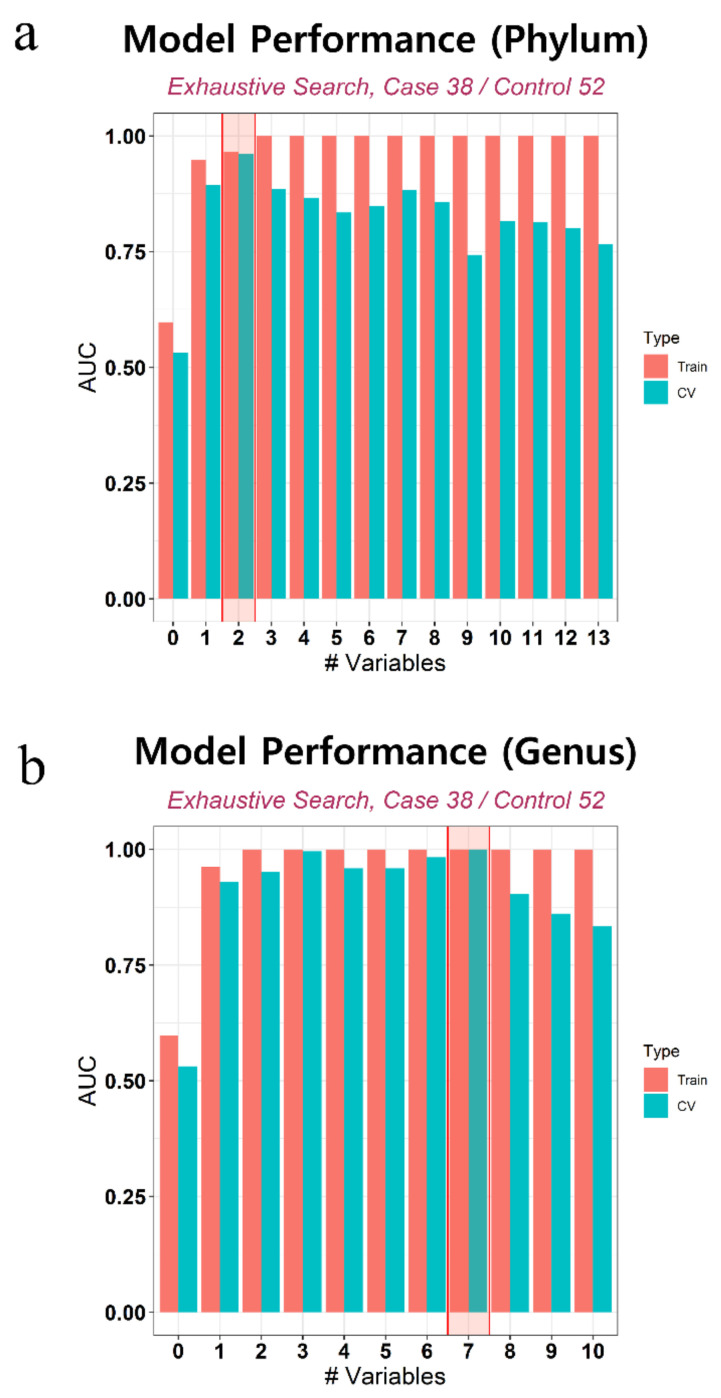
The bar plot describes the overall performance of models at phylum and genus level.

**Figure 5 biology-10-00219-f005:**
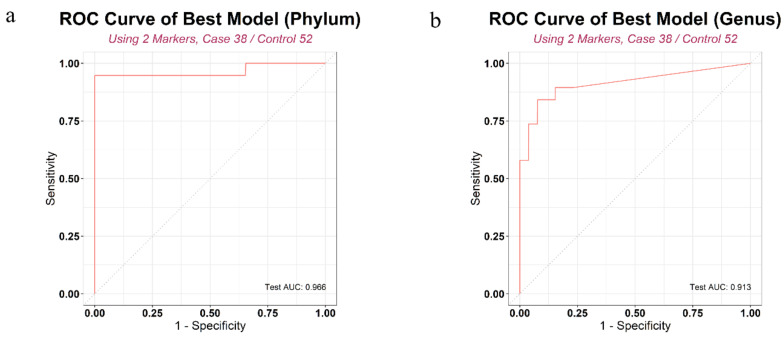
ROC curve of prediction model: (**a**) phylum level, (**b**) genus level. Logistic regression model was built using microbiome markers and covariates (age, gender) to distinguish PC and control groups.

**Figure 6 biology-10-00219-f006:**
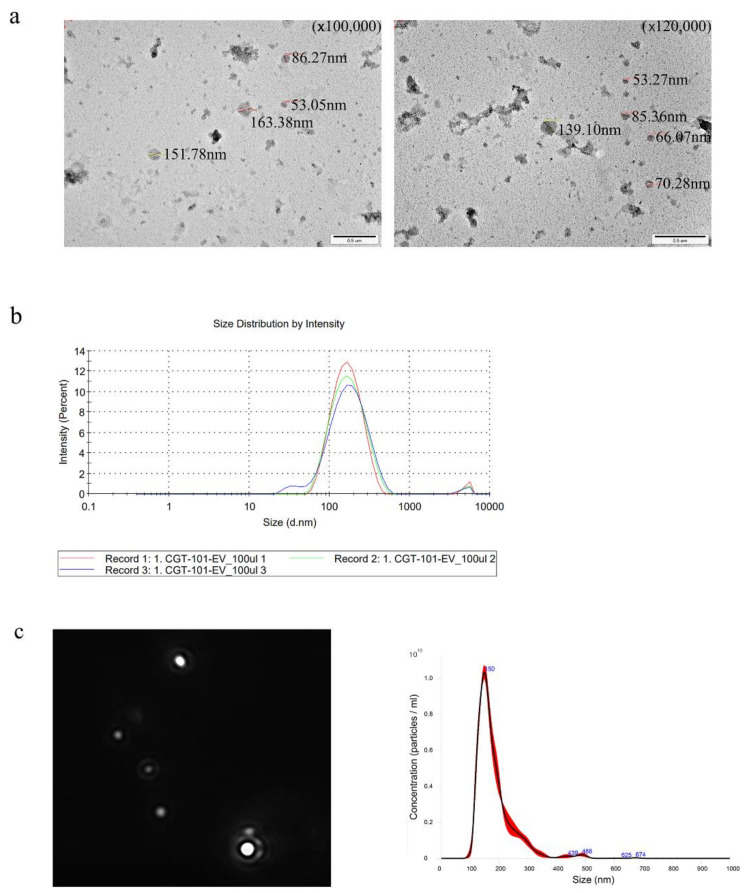
Biologic characteristics of *C. glutamicum* EV through in vitro experiment: (**a**) TEM, (**b**) Dynamic Light Scattering (DLS), (**c**) Nanoparticle Tracking Analysis (NTA), (**d**) Sodium Dodecyl Sulfate-PolyAcrylamide Gel Electrophoresis (SDS-PAGE), (**e**) The amount of TNF-α secretion according to the concentration of C. glutamicum EV, using E. coli EV as stimulant.

**Table 1 biology-10-00219-t001:** Basic characteristics of patients.

**(A)** Age and sex of entire patients before and after PSM									
		**Before PSM (n = 198)**				**After PSM (n = 90)**			
		**Pancreatic Cancer (n = 82)**		**Controls (n = 116)**	***p*-Value ***	**Pancreatic Cancer (n = 38)**		**Controls (n = 52)**	***p*-Value ***
**Sex**	Male	55 (67.1%)		19 (16.4%)	1.131 × 10^−12^	17 (44.7%)		17 (32.7%)	0.35
	Female	27 (32.9%)		97 (83.6%)		21 (55.3%)		35 (67.3%)	
Age (mean ± SD)		63.07 ± 9.83		49.28 ± 12.45	1.4 × 10^−12^	57.24 ± 8.38		57.63 ± 10.50	0.51
**(B)** Clinicopathologic characteristics of patients with pancreatic cancer after PSM									
							**Pancreatic Cancer (n = 38)**		
Age, Mean ± SD							57.2 ± 8.4		
Sex, M:F							17:21		
CEA > 5							5 (13.2%)		
CA 19-9 > 37							30 (78.9%)		
Neoadjuvant chemotherapy							7 (18.4%)		
Neoadjuvant radiotherapy							5 (13.2%)		
Operation			R0				26 (68.4%)		
			R1				6 (15.8%)		
			R2				6 (15.8%)		
Size, mean ± SD							3.9 ± 1.7		
Histology			Adenocarcinoma				36 (94.7%)		
			Others				2 (5.3%)		
Stage (AJCC stage, 7th)			I				1 (2.63%)		
			II				26 (68.4%)		
			III				5 (13.2%)		
			IV				6 (15.8%)		

(A): PSM: propensity score matching, SD: standard deviation. * Chi-square test or Wilcoxon rank-sum test. (B): SD: standard deviation, CEA: carcinoembryonic antigen, CA 19-9: carbohydrate antigen 19-9, AJCC: American Joint Committee on Cancer.

**Table 2 biology-10-00219-t002:** Selected biomarkers at phylum and genus level using 9 statistical methods. Markers were organized in the order that were significant in many methods.

(A) Phylum (L2) level											
Phylum	CLR_Perm	DESeq2_LRT	DESeq2_Wald	edgeR	Wilcoxon	ZIBSeq	ZIG_Gaussian	ZIG_log_Normal	ANCOM	Freq	Sig.
*Verrucomicrobia*	0.0000	0.0000	0.0000	0.0010	0.0000	0.0000	0.0000	0.0000	Verrucomicrobia	case	9
*Deferribacteres*	0.0000	0.0000	0.0000	0.2083	0.0019	0.5871	0.0000	0.0343	Deferribacteres	case	7
*Actinobacteria*	0.0000	0.0016	0.0011	0.0512	0.0204	0.0001	0.0057	0.5981	Actinobacteria	control	7
*Bacteroidetes*	0.0588	0.0210	0.0222	0.8176	0.0317	0.0021	0.0087	0.8280	Bacteroidetes	case	6
*SR1 (Absconditabacteria)*	0.7637	0.0000	0.0000	0.8176	0.4562	0.9999	0.0379	0.7192	-	control	3
*Spirochaetae*	0.1657	0.0022	0.0018	1.0000	0.5852	0.9999	0.0000	0.7192	-	control	3
*Proteobacteria*	0.0525	0.0032	0.0023	0.2005	0.1705	0.0013	0.0872	0.7192	-	control	3
*Planctomycetes*	0.2538	0.0026	0.0021	1.0000	0.5852	0.9999	0.0000	0.7192	-	control	3
*FBP*	0.1628	0.0019	0.0015	1.0000	0.5852	0.9999	0.0000	0.7192	-	control	3
*Cyanobacteria*	0.2538	0.0001	0.0000	0.1059	0.4562	0.5871	0.0010	0.5981	-	control	3
*Chloroflexi*	0.2682	0.0002	0.0001	1.0000	0.9513	0.0763	0.0000	0.7192	-	control	3
*Armatimonadetes*	0.1050	0.0001	0.0000	1.0000	0.1153	0.9999	0.0005	0.7192	-	control	3
*Acidobacteria*	0.7637	0.0009	0.0011	1.0000	0.4562	0.9999	0.0379	0.7192	-	control	3
**(B) Genus (L6) level**											
**Genus**	**CLR_Perm**	**DESeq2_LRT**	**DESeq2_Wald**	**edgeR**	**Wilcoxon**	**ZIBSeq**	**ZIG_Gaussian**	**ZIG_log_Normal**	**ANCOM**	**Freq**	**Sig.**
*Stenotrophomonas*	0.0159	0.0000	0.0000	1.0000	0.0095	0.0028	0.0000	8.82474E-06	Significant	control	8
*Sphingomonas*	0.0000	0.0002	0.0000	1.0000	0.0042	0.0020	0.0000	0.000394665	Significant	control	8
*Ruminococcaceae UCG-014*	0.0159	0.0000	0.0000	1.0000	0.0006	0.0000	0.0001	4.27329E-07	Significant	case	8
*Propionibacterium*	0.0000	0.0000	0.0000	1.0000	0.0004	0.0251	0.0000	0.000131155	Significant	control	8
*Lachnospiraceae NK4A136 group*	0.0000	0.0032	0.0030	1.0000	0.0043	0.0001	0.0009	0.000453009	Significant	case	8
*Akkermansia*	0.0000	0.0000	0.0000	1.0000	0.0000	0.0000	0.0000	0.278837499	Significant	case	7
*Turicibacter*	0.0000	0.0000	0.0000	1.0000	0.0052	0.7879	0.0109	0.043444196	-	case	6
*Ruminiclostridium*	0.0000	0.0015	0.0011	1.0000	0.0306	0.1365	0.0000	0.049630223	-	case	6
*Lachnospiraceae UCG-001*	0.0422	0.0000	0.0000	1.0000	0.0361	0.9999	0.0004	0.043444196	-	case	6
*Corynebacterium 1*	0.0159	0.0003	0.0000	1.0000	0.0131	0.8506	0.0126	0.159296954	Significant	control	6

## Data Availability

The data presented in this study are available on request from the corresponding author. The data are not publicly available due to patients’ privacy.

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
