# Peer review of "Microbiome Markers of Pancreatic Cancer Based on Bacteria-Derived Extracellular Vesicles Acquired from Blood Samples: A Retrospective Propensity Score Matching Analysis"

_biology, 2021, doi:10.3390/biology10030219_

Round 1

Reviewer 1 Report

The authors adequately address my initial review critiques and I appreciate their detail with the rebuttal.

Reviewer 2 Report

The paper entitled: "Microbiome markers of pancreatic cancer based on bacteria-derived extracellular vesicles acquired from blood samples: A retrospective propensity score matching analysis" by Jae Ri Kim et al, is now corret.

This manuscript is a resubmission of an earlier submission. The following is a list of the peer review reports and author responses from that submission.

Round 1

Reviewer 1 Report

The blood of pancreatic cancer patients may hold novel biomarkers of disease progression. Accordingly, this study investigates bacterial-derived extracellular vesicles (EVs) in the blood from 38 patients and 52 healthy controls using 16S rRNA analysis and further examining bacteria-derived EVs. The study reports differences in microbial compositions at both the phylum and genus levels in PC patients. A PC prediction model based on microbial marker combinations demonstrated specificity of candidate microbiome markers in patients with PC.

This study presents some interesting and novel findings with respect to circulating biomarkers derived from 16S rRNA analysis. However, there are some major problems with the description and further analysis is needed to publish this manuscript in Biology.

  1. Metagenomics is the study of the functional genomes of microbial communities while 16S sequencing provides a phylogenetic survey on the diversity of a single ribosomal gene. Therefore, the term metagenomics is incorrectly used throughout and it is necessary to explicitly state this as ‘16S rRNA’. Otherwise, the study needs to include whole genome or shotgun metagenomics sequencing to use the term metagenome.
  2. Propensity score matching is commonly be used to equate treatment groups with respect to measured baseline covariates to achieve a comparison with reduced selection bias. Generally, PSM requires large samples, with substantial overlap between treatment and control groups. However, the authors fail to provide any indication that this method is suitable for this study (i.e., its feasibility with regard to the data and any potential confounders, what are the set of propensity scores from a logistic regression model with treatment group as the outcome and the balancing factors as predictors? Match patients in the treatment groups with similar propensity scores, balancing all factors? To assess the success of the matching with balance diagnostics, graphically or analytically?). The information presented in the appendix is unreadable and blurry. The legends provide little detail and is uninterpretable as is.
  3. It is unclear that the in vitro characterization analysis is relevant to the study. First, this is not appropriately rationalized, described or presented in the results section. The authors examine EVs from cultures, but it is implied that these results are meant to characterize the EVs. I see no controls for the microscopy characterization helping to point out that bacterial DNA is contained within these vesicles and whether this is meaningful towards a biological conclusion regarding PC blood biomarkers. It would better to characterize EV from blood with controls. Also, what the relevance is of the protein content via SDS PAGE and protein staining?
  4. The details of the model are virtually lacking and need to be described and presented.
  5. Figure 1 is blurry and incomprehensible.
  6. Figure 4 legend and results text lacks enough detail to fully interpret the results.
  7. Table 3 should be presented differently or made supplemental.
  8. Did the authors find evidence of Helicobacter pylori and other PC associated microorganisms?

Reviewer 2 Report

Jae Ri Kim et al, et in the report entitled: "Metagenomic markers of pancreatic cancer based on bacteria-derived extracellular vesicles acquired from blood samples: A retrospective propensity score matching analysis" report microbiome markers that may be candidate for early diagnosi of pancreatic cancer. I suggest to shorten the discussion section.